# Extensive Mendelian randomization study identifies potential causal risk factors for severe COVID-19

Yitang Sun [1], Jingqi Zhou[1,2] & Kaixiong Ye [1,3✉]

## Abstract

**Background** Identifying causal risk factors for severe coronavirus disease 2019 (COVID-19) is critical for its prevention and treatment. Many associated pre-existing conditions and biomarkers have been reported, but these observational associations suffer from confounding and reverse causation.

**Methods** Here, we perform a large-scale two-sample Mendelian randomization (MR) analysis to evaluate the causal roles of many traits in severe COVID-19.

**Results** Our results highlight multiple body mass index (BMI)-related traits as risk-increasing: BMI (OR: 1.89, 95% CI: 1.51–2.37), hip circumference (OR: 1.46, 1.15–1.85), and waist circumference (OR: 1.82, 1.36–2.43). Our multivariable MR analysis further suggests that the BMI-related effect might be driven by fat mass (OR: 1.63, 1.03–2.58), but not fat-free mass (OR: 1.00, 0.61–1.66). Several white blood cell counts are negatively associated with severe COVID-19, including those of neutrophils (OR: 0.76, 0.61–0.94), granulocytes (OR: 0.75, 0.601–0.93), and myeloid white blood cells (OR: 0.77, 0.62–0.96). Furthermore, some circulating proteins are associated with an increased risk of (e.g., zinc-alpha-2-glycoprotein) or protection from severe COVID-19 (e.g., prostate-associated microseminoprotein).

**Conclusions** Our study suggests that fat mass and white blood cells might be involved in the development of severe COVID-19. It also prioritizes potential risk and protective factors that might serve as drug targets and guide the effective protection of high-risk individuals.

## Plain language summary

People infected with SARS-CoV-2 can remain asymptomatic, have mild symptoms, or develop severe COVID-19 that increases their risk of death. Finding factors that directly contribute to the risk of developing severe COVID-19, so-called causal risk factors, may help in prevention by identifying individuals at higher risk and in treatment by providing clues of targets for therapies. We applied advanced statistical methods that leverage genetic variations among individuals to separate causal risk factors from coincidences, sifting through tens of thousands of candidate factors. We show that levels of fat mass, certain white blood cells, and multiple circulating proteins are potential causal risk factors for severe COVID-19. These findings help us better understand severe COVID-19 and guide future studies to develop strategies of prevention and treatment.

[1] Department of Genetics, Franklin College of Arts and Sciences, University of Georgia, Athens, GA, USA. [2] School of Public Health, Shanghai Jiao Tong University School of Medicine, Shanghai, China. [3] Institute of Bioinformatics, University of Georgia, Athens, GA, USA. ✉email: Kaixiong.Ye@uga.edu

The coronavirus disease 2019 (COVID-19) is a global pandemic caused by severe acute respiratory syndrome coronavirus 2 (SARS–CoV–2)[1]. As of mid-April 2021, 146 million confirmed cases and three million deaths from COVID-19 have been reported worldwide[2]. Despite substantial public health and medical efforts, COVID-19 continues to cause irreversible damage and death[3–5]. It is essential to identify risk factors and potential drug targets for COVID-19 in order to improve primary prevention and to develop treatment strategies.

Many observational studies have reported that older age, male gender, non-White ethnicity, and pre-existing conditions, such as cardiovascular disease, diabetes, chronic respiratory disease, hypertension, and cancers, are associated with increased COVID-19 susceptibility and severity[5–8]. Moreover, retrospective observational studies have noted that hospitalized COVID-19 patients, especially those with severe respiratory or systemic conditions, are at increased risks of atrial fibrillation, nonsustained ventricular tachycardia, acute kidney injury, neurologic disorders, and thrombotic complications[9–12]. Vitamin-D deficiency, higher body mass index (BMI), and obesity have been associated with an increased risk of COVID-19[13,14]. Some lifestyle factors were also identified as risk-increasing, such as smoking, alcohol consumption, and lack of physical activity[15]. However, it is difficult to infer causal effects from observational studies because they are susceptible to confounding and reverse causation, while data from randomized controlled trials are scarce and inconclusive.

Mendelian randomization (MR) study provides a promising opportunity to validate and prioritize putative risk factors and drug targets. MR studies use randomly allocated genetic variants related to the exposure as instrumental variables for investigating the effect of the exposure on an outcome[16]. It is expected to be independent of confounding factors and has been demonstrated as an efficient and cost-effective strategy to identify causal effects[17]. Recent MR studies have provided evidence of causality for a range of risk factors on COVID-19 (Supplementary Data 1). For instance, BMI and smoking are associated with an increased COVID-19 risk, while no evidence of causal effects was found for circulating 25-hydroxy-vitamin-D levels[18–20]. However, inconsistent results were also reported for some factors, such as Alzheimer's disease, blood lipids, and physical activity[18,19,21–26]. Some of these inconsistencies are likely due to the usage of early genome-wide association studies (GWAS) of COVID-19, which have small sample sizes. Moreover, most studies are limited to a small number of candidate factors, leaving many more to be tested and identified. The recent release of large GWAS meta-analysis for various COVID-19 phenotypes offers a great opportunity for MR studies[27]. However, special care and caution are also needed when interpreting the results. Sampling from COVID-19 patients, individuals tested for infection, voluntary participants, or existing cohorts may result in nonrepresentative samples and induce collider bias that distorts phenotypic and genetic associations[28–30]. The inherent complexity of COVID-19 as an infectious disease and the potential complications in ascertaining cases and controls make it challenging to disentangle risk factors for an increased chance of infection, susceptibility to infection, and disease severity[27,28].

In this study, we conducted an unbiased and large-scale MR analysis to examine the potential causal effects of an extensive list of exposures on severe COVID-19. All existing GWAS, as compiled by the Integrative Epidemiology Unit (IEU) OpenGWAS project, were included[31,32]. We note that some GWAS were on the same traits. In each of these GWAS, independent genetic variants at the genome-wide significance were selected as instrumental variables for the studied trait. The associations between genetic instruments and the risk of severe COVID-19 were evaluated based primarily on three nonindependent GWAS

of COVID-19. The COVID-19 Host Genetics Initiative (HGI) study A2 from release 4 alpha was used in our discovery analysis. HGI A2 compared COVID-19 patients with confirmed severe respiratory symptoms to population controls[27]. The HGI B2 study, comparing hospitalized COVID-19 patients to population controls, was used as one of our two replication datasets. The other replication dataset, labeled as the NEJM study, was drawn from the first published GWAS study of COVID-19 comparing patients with respiratory failure to healthy controls from Italy and Spain[33]. We note that due to sample overlap and different phenotypic definitions, the HGI B2 and NEJM studies are not independent or strict replications of HGI A2. They mainly serve the purpose of reducing false positives in our prioritized list of risk factors. To ensure the robustness of the prioritized list of risk factors, results based on these three COVID-19 GWAS were compared to those from different releases (4 alpha vs. 4 and 5), different case definitions (very severe respiratory COVID-19 in A2 and hospitalized COVID-19 in B2 vs. any reported infection in C2), and different control groups (population controls in A2 and B2 vs. nonhospitalized COVID-19 patients in A1 and B1). Furthermore, multiple sensitivity analyses were performed to detect and correct for the presence of pleiotropy in genetic instruments. Here, we only report associations that do not have evidence of pleiotropy in genetic instruments and are observed in at least one of the two primary replication analyses. As an in-depth investigation into the BMI-related traits, we further conducted a multivariable MR analysis to disentangle the effects of fat mass and fat-free mass. Our findings provide profound insights into the etiology of severe COVID-19 and prioritize candidate causal risk factors for public health intervention and for drug discovery.

## Methods

**Exposure data sources**. This study analyzed publicly available summary statistics from previous GWAS and did not include individual-level data. Ethical approval or informed consent was not required.

To obtain a comprehensive list of traits with existing GWAS, the summary statistics of 34,519 published GWAS were extracted from the MRC Integrative Epidemiology Unit (University of Bristol) GWAS database (https://gwas.mrcieu.ac.uk/). Details of each GWAS study can be found at https://gwas.mrcieu.ac.uk/datasets/. The R package TwoSampleMR (version 0.5.5) was applied to retrieve the IEU GWAS datasets[31,32]. The univariable MR study was conducted using the same package. This study is reported as per the guidelines for strengthening the reporting of Mendelian randomization studies (STROBE-MR, Supplementary Data 2)[34].

These GWAS were further filtered based on the following criteria: (1) European ancestry; (2) not eQTL studies, those labeled as "eqtl" from eQTLGen 2019[35]. A total of 14,385 GWAS summary datasets were retained and used in this study. Detailed information on data sources, all GWAS, and their corresponding traits are available in Supplementary Data 3. The units of the exposures follow the definitions in the prior GWAS since we directly used the existing summary statistics. We note that all exposures reported in the main text and Supplementary Data 4 have the unit of standard deviation (SD).

**Outcome data sources**. For evaluation of the association with COVID-19 severity, the instrument-outcome effects were retrieved from the recent version of GWAS meta-analysis by the COVID-19 Host Genetics Initiative (HGI, release 4 alpha, accessed on October 9, 2020)[27]. Detailed information has been provided on the COVID-19 HGI website (https://

www.covid19hg.org/results/). In our primary discovery analysis, we used the summary statistics based on the comparison of 2972 patients confirmed as "very severe respiratory" COVID-19 with the 284,472 general population samples. This is called "the HGI A2 study".

To reduce false positives and to ensure the robustness of our discoveries, replication analyses were performed with two additional GWAS of COVID-19. One of them was also from the COVID-19 HGI, comparing 6492 hospitalized COVID-19 patients with 1,012,809 control participants. We called it "the HGI B2 study". Only single nucleotide polymorphisms (SNPs) with imputation-quality scores >0.6 were retained. The other GWAS was on 1610 COVID-19 patients with respiratory failure and 2180 controls from Italy and Spain, and it was called "the NEJM study"[33].

To evaluate the robustness of our findings to different data releases, we performed additional analyses with the A2 and B2 COVID-19 GWAS from HGI releases 4 and 5. Since the A2 and B2 GWAS utilized population samples with unknown COVID-19 status, we further performed MR analyses with the A1 and B1 COVID-19 GWAS, which utilized nonhospitalized COVID-19 patients as the control, in order to evaluate the impact of different control samples. In an attempt to distinguish the effects of risk factors on COVID-19 susceptibility and severity, we also included HGI C2 studies, which compared any COVID-19 case (laboratory-confirmed or clinically confirmed SARS-CoV-2 infection, or self-reported COVID-19) to population controls. Summary statistics from HGI releases 4 and 5 were accessed on March 27, 2021 and analyzed with the same computational pipeline as described before. In addition, genetic correlations between various COVID-19 phenotypes from HGI releases 4 and 5 were estimated using linkage-disequilibrium score (LDSC) regression[36,37].

**Selection of instrumental variables**. For the implementation of MR, SNPs were selected based on the genome-wide significance threshold ($p < 5 \times 10^{-8}$). To ensure that SNPs are independent, we pruned the variants by linkage disequilibrium (LD) ($R^2$ threshold of 0.001 or clumping window of 10,000 kb). When target SNPs were not present in the outcome dataset, proxy SNPs were used instead through LD tagging (minimum LD $R^2$ threshold of 0.8). The effect alleles of selected genetic variants were harmonized across the exposure and outcome associations. F statistics were calculated to assess instrument strength[38]. F statistics $\geq 10$ indicate strong instruments.

**Univariable Mendelian randomization**. Two-sample Mendelian randomization analysis was undertaken using GWAS summary statistics for each exposure-outcome pair. In order to estimate the causal effect of each trait on severe COVID-19, the inverse-variance-weighted (IVW) method with a multiplicative random-effect model was used as the primary analysis[39–41]. Horizontal pleiotropy occurs when SNPs exert an effect on severe COVID-19 through other biological pathways independent of the studied exposure. To assess the presence of heterogeneity among genetic instruments, Cochran's Q statistic was calculated for heterogeneity for the IVW analyses[42]. An extended version of Cochran's Q statistic (Rücker's Q′) can be estimated for the MR-Egger[43]. We used the MR-Egger intercept test to evaluate the presence of unbalanced horizontal pleiotropy[40]. To account for pleiotropy, additional sensitivity analyses were performed with the MR-Egger[40,41], weighted median (WM)[44], and weighted-mode methods[45]. The MR-Egger method allows unbalanced horizontal pleiotropic effects even when all SNPs are invalid instruments[40]. The WM method can provide robust causal estimates when at least 50% of SNPs are valid genetic instruments, while the weighted-mode method reports the causal-effect estimate supported by the largest number of instruments[44,45]. The false-discovery rate (FDR) approach was utilized to correct for multiple testing, and it was applied to the $p$ values from the IVW random-effect model[46]. An association was declared significant if the q-value is < 0.05, and was deemed suggestive if the unadjusted $p$-value is < 0.05.

Two additional exclusion criteria were applied to filter out exposures before FDR correction: (1) the number of genetic instruments was less than three. Three or more are required for statistical tests of pleiotropic effects and for statistical sensitivity analyses to correct for pleiotropy. (2) Exposures with indications of pleiotropy in their genetic instruments. The presence of pleiotropy violates the assumption of MR analysis. For the remaining exposures, FDR correction for multiple testing was applied separately for each analysis with the HGI A2, HGI B2, or NEJM study. To identify potential causal risk factors for severe COVID-19, we used two approaches to consider the evidence strength. First, the significant and replicated results were defined as those with a $q$-value < 0.05 in the discovery analysis and a $p$-value < 0.05 in either one of the replication studies (Supplementary Data 5). Second, the suggestive and replicated results were defined as those with a $p$-value < 0.05 in the discovery analysis and a $p$-value < 0.05 in either one of the replication studies (Supplementary Data 6). All MR analyses were conducted in R with the TwoSampleMR package[31]. Additional MR analyses were also performed using GWAS from HGI releases 4 and 5. An analysis flowchart is shown in Fig. 1.

For a few exposures that have a small number of genetic instruments, we performed an exemplary sensitivity analysis by excluding SNPs with potential pleiotropic effects. For each SNP, we queried the PhenoScanner[47,48] and retrieved any associations at the genome-wide significance. After excluding SNPs with associations with blood cell or BMI-related traits, we repeated the MR analysis as described before.

**Multivariable Mendelian randomization**. As many BMI-related traits are typically correlated with each other, we conducted a two-sample multivariable MR (MVMR) analysis to explore independent causal risk factors for severe COVID-19[49]. SNPs associated with fat mass and fat-free mass were obtained from previous GWAS by MRC IEU and the Neale Lab through the TwoSampleMR package. The effects of genetically predicted fat mass and fat-free mass for each pair of the whole body, left arm, right arm, left leg, right leg, and trunk were estimated using the MVMR package (version 0.2.0) in R.

**Reporting summary**. Further information on research design is available in the Nature Research Reporting Summary linked to this article.

## Results
**Study overview**. The workflow of our study is summarized in Fig. 1. Starting with the 34,519 GWAS compiled by the IEU OpenGWAS project, we focused on the 14,385 GWAS that are based on European-descent samples, in order to match the major ancestry in the GWAS of COVID-19 and to avoid false positives as results of population discrepancy in genetic effects. The details of traits and their GWAS included in our study are provided in Supplementary Data 3. From each GWAS, genetic instruments were selected as independent genetic variants at the genome-wide significance. Three or more genetic instruments are required for statistical tests of pleiotropic effects, and thus exposures with fewer instruments were excluded. For the univariable MR analysis

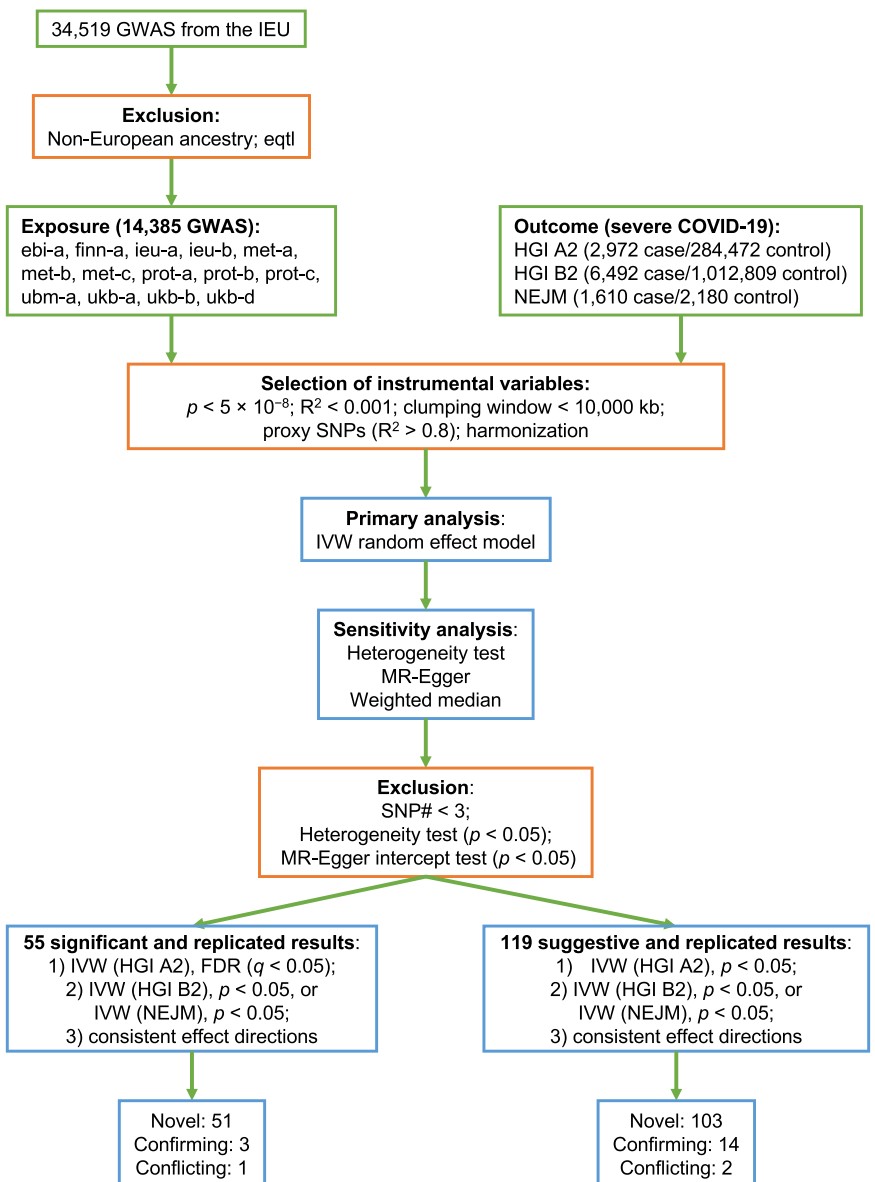

**Fig. 1 The workflow of our extensive MR study for severe COVID-19.** GWAS genome-wide association studies, IEU Integrative Epidemiology Unit, eqtl expression quantitative trait loci, HGI host genetics initiative, NEJM New England journal of medicine, IVW inverse-variance weighted, SNP single-nucleotide polymorphism, SNP# number of SNPs used as genetic instruments, Novel not reported before, Confirming confirming some previously reported results, even though previous results may be conflicting among themselves; Conflicting conflicting with previously reported results. The definition of novelty is by comparison to existing COVID-19 MR studies, as summarized in Supplementary Data 1. Detailed summary statistics could be found in Supplementary Data 5 and 6.

of each exposure-outcome pair, we first applied the IVW method with a multiplicative random-effect model[39]. We then evaluated the possible presence of pleiotropic effects with Cochran's Q test of heterogeneity and the MR-Egger intercept test for directional pleiotropy[40,42,43]. We excluded all exposures with indications of pleiotropy in their genetic instruments to fulfill the key assumptions underlying MR analysis. We retained 1817 exposure GWAS for the discovery analysis with the HGI A2 study, 1740 for the replication analysis with the HGI B2 study, and 1915 for the replication analysis with the NEJM study (Supplementary Data 7). All retained traits have F statistics ≥10, indicating strong genetic instruments. The FDR approach was utilized in each MR analysis to correct for multiple testing of many exposures and to reduce false positives. Based on these three sets of analysis, we defined two sets of results: (1) the significant and replicated results, which have a $q$-value < 0.05 in the discovery analysis and

a $p$-value < 0.05 in either one of the replication studies (Supplementary Data 5); and (2) the suggestive and replicated results, which have a $p$-value < 0.05 in the discovery analysis and a $p$-value < 0.05 in either one of the replication studies (Supplementary Data 6). A total of 55 significant and replicated traits were identified. Among them, 17 were replicated in both replication datasets (Supplementary Data 4 and Supplementary Figs. 1–3).

**BMI-related traits**. In the univariable MR study, eight BMI-related traits are positively associated with severe COVID-19 in our discovery analysis and also in both of our replication analyses (Supplementary Data 4). Genetically predicted one SD-increase of BMI is associated with a higher risk of severe COVID-19 (OR per SD increment: 1.89, 95% CI: 1.51–2.37, $p = 3.15 \times 10^{-8}$)

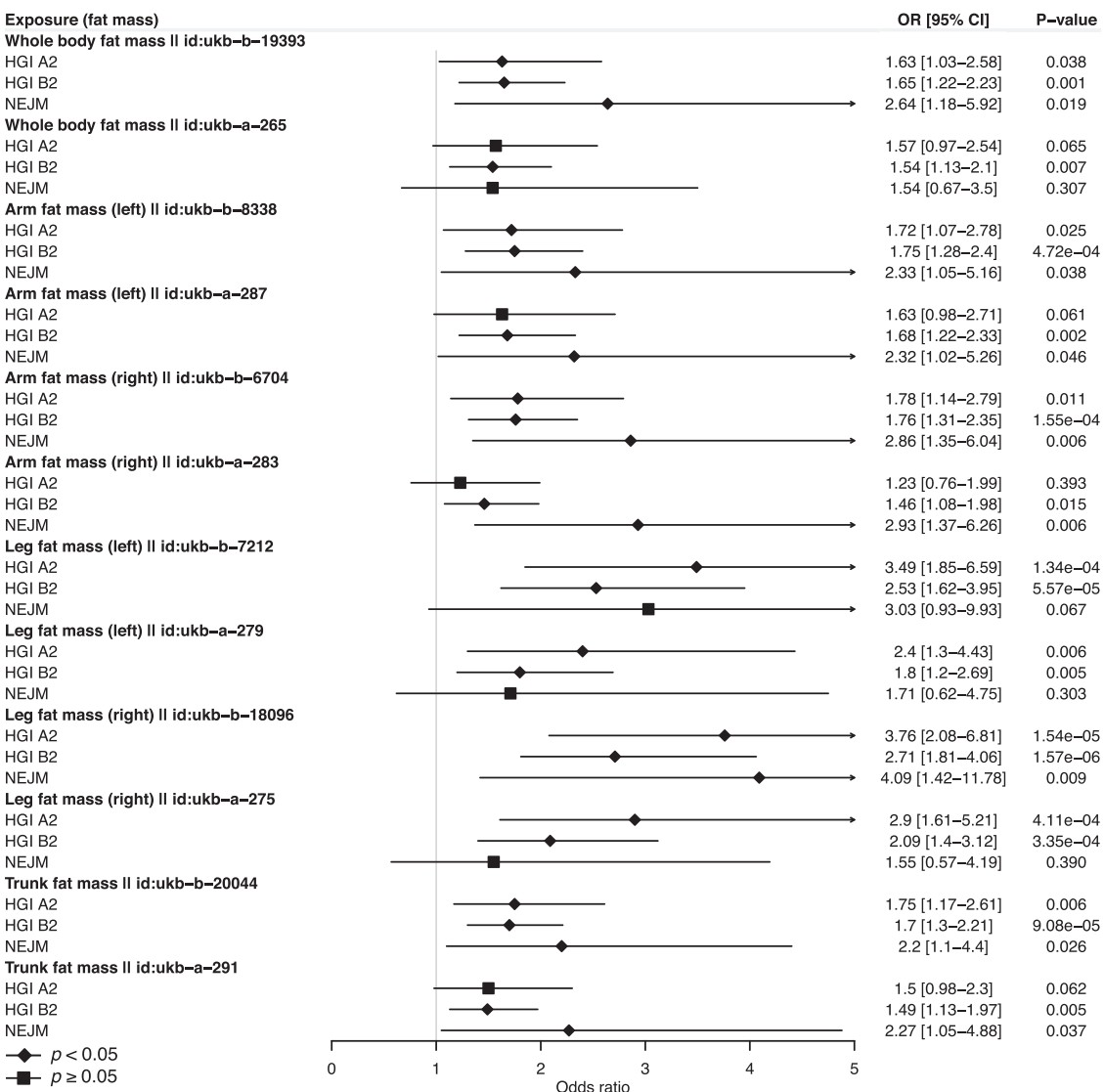

**Fig. 2 Multivariable MR analysis reveals support for independent causal roles of fat mass in severe COVID-19.** The effects of fat mass at different body parts were shown. Each specific fat mass has two GWAS sources, whose ID in the Integrative Epidemiology Unit (IEU) OpenGWAS project was shown right after the exposure name. Odds ratios (OR) and 95% confidence intervals (CI) are scaled to a genetically predicted 1-standard-deviation (SD) increase in fat mass. Three COVID-19 GWAS were shown. HGI A2 host genetics initiative study A2, HGI B2 host genetics initiative study B2, NEJM the study published in New England Journal of Medicine. Associations with p-value < 0.05 were indicated with diamonds, while others with squares. Error bars stand for 95% CI. Detailed summary statistics could be found in supplementary Data 8.

(Supplementary Fig. 1). Consistent with the effect of BMI, genetically instrumented higher hip circumference (OR: 1.46, 95% CI: 1.15–1.85, $p = 0.0017$) and waist circumference (OR: 1.82, 95% CI: 1.36–2.43, $p = 6.20 \times 10^{-5}$) are associated with a higher risk. The univariable MR study also provided strong evidence that weight and fat mass in the left arm, right arm, left leg, right leg, trunk, and whole body are positively associated with severe COVID-19 (Supplementary Data 6).

To pinpoint the different aspects of BMI-related traits, we investigated the roles of fat mass and fat-free mass indices in severe COVID-19 (Supplementary Data 8). In the multivariable MR analysis controlling for fat-free mass, there is strong evidence for direct causal effects of fat mass measured at different body parts, including the whole body, left and right arms, left and right legs, and the trunk. The evidence is consistent across the three GWAS of COVID-19 severity (Fig. 2). On the other hand, there is no evidence for direct causal effects of fat-free mass (Fig. 3). The multivariable MR

analysis results suggest that the causal effects of BMI-related traits on severe COVID-19 might be mainly driven by fat mass.

**White blood cell traits**. In the univariable MR analyses, we identified a group of five white blood cell traits to be negatively associated with the risk of severe COVID-19. Specifically, suggestive associations were determined for neutrophil count (OR per SD increment: 0.76, 95% CI: 0.61–0.94, $p = 0.013$), sum basophil–neutrophil counts (OR: 0.71, 95% CI: 0.57–0.87, $p = 0.001$), sum neutrophil–eosinophil counts (OR: 0.76, 95% CI: 0.61–0.95, $p = 0.015$), myeloid white cell count (OR: 0.77, 95% CI: 0.62–0.96, $p = 0.0197$), and granulocyte count (OR: 0.75, 95% CI: 0.601–0.93, $p = 0.009$) (Fig. 4). For all five traits, causal estimates are broadly concordant in WM and weighted-mode methods, and consistent directions of the effects were also found by the MR-Egger method (Supplementary Data 6). Take neutrophil count as an example, consistent estimates of a protective effect were found with WM (OR: 0.61, 95% CI: 0.42–0.88, $p = 0.009$) and weighted mode (OR: 0.59, 95% CI: 0.39–0.91, $p = 0.017$).

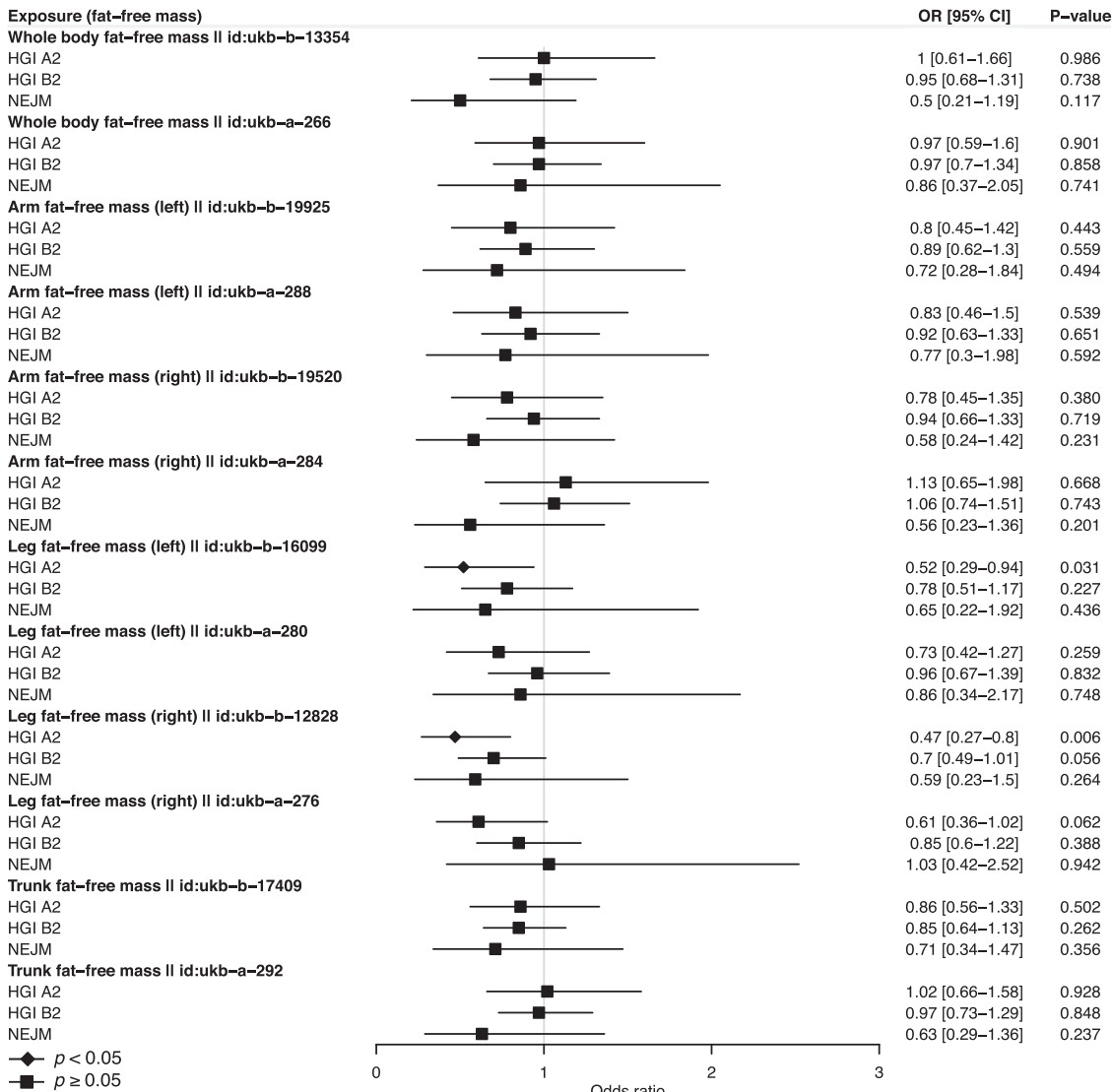

**Fig. 3 Multivariable MR analysis reveals no support for causal roles of fat-free mass in severe COVID-19.** The effects of fat-free mass at different body parts were shown. Each specific fat-free mass has two GWAS sources, whose ID in the Integrative Epidemiology Unit (IEU) OpenGWAS project was shown right after the exposure name. Odds ratios (OR) and 95% confidence intervals (CI) are scaled to a genetically predicted 1-standard-deviation (SD) increase in fat mass. Three COVID-19 GWAS were shown. HGI A2 host genetics initiative study A2, HGI B2 host genetics initiative study B2, NEJM the study published in New England Journal of Medicine. Associations with $p$-value < 0.05 were indicated with diamonds, while others with squares. Error bars stand for 95% CI. Detailed summary statistics could be found in Supplementary Data 8.

Overall, our findings support the potential causal roles of white blood cells, especially neutrophils, in reducing the risk of developing severe COVID-19.

**Circulating proteins.** Our univariable MR analyses revealed evidence of potential causal effects for some circulating proteins. There are five proteins whose effects on severe COVID-19 are significant in the discovery MR analysis (q-value < 0.05) and also replicated in both replication analyses (p-value < 0.05) (Supplementary Data 4 and Supplementary Fig. 2). Two of them are negatively associated with the risk of severe COVID-19, including interleukin-3-receptor subunit alpha (OR per SD increment: 0.87, 95% CI: 0.79–0.94) and prostate-associated microseminoprotein (OR: 0.71, 95% CI: 0.58–0.86). The other three are risk-increasing, including zinc-alpha-2-glycoprotein (OR: 1.37, 95% CI: 1.14–1.66), C1GALT1-specific chaperone 1 (OR: 1.20, 95% CI: 1.19–1.21), and corneodesmosin (OR: 1.12, 95% CI: 1.09–1.16).

There are another six circulating proteins that have significant and replicated effects on severe COVID-19, although they are only replicated in one replication analysis (Supplementary Data 5): inter-alpha-trypsin inhibitor heavy chain H1 (OR: 1.08, 95% CI: 1.04–1.12), alpha-2-macroglobulin receptor-associated protein (OR: 1.14, 95% CI: 1.07–1.23), resistin (OR: 1.09, 95% CI: 1.07–1.11), reticulon-4 receptor (OR: 0.86, 95% CI: 0.79–0.93), C–C motif chemokine 23 (OR: 0.88, 95% CI: 0.83–0.92), and collectin-10 (OR: 0.83, 95% CI: 0.76–0.901). Additionally, our suggestive and replicated results revealed another 13 proteins to be associated with the severe COVID-19 risk (Supplementary Data 6). Overall, our MR analyses prioritized scores of circulating proteins that are potentially causal in the development of severe COVID-19.

**Comparisons across HGI releases and phenotype definitions.** While our primary analysis utilized COVID-19 GWAS (A2 and

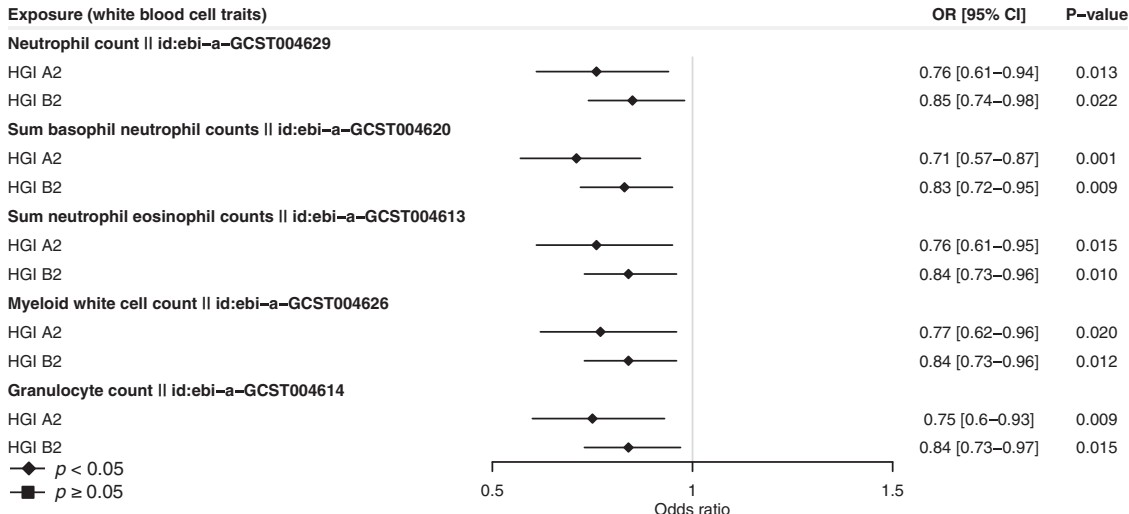

**Fig. 4 MR analysis of white blood cell traits on severe COVID-19 risk.** The effects of counts of different white blood cells were shown. For each specific blood-cell count, its GWAS ID in the Integrative Epidemiology Unit (IEU) OpenGWAS project was shown right after the exposure name. Odds ratios (OR) and 95% confidence intervals (CI) are scaled to a genetically predicted 1-standard-deviation (SD) increase in white blood cell trait. Two COVID-19 GWAS were shown. HGI A2 host genetics initiative study A2, HGI B2 host genetics initiative study B2. Associations with $p$-value < 0.05 were indicated with diamonds, while others with squares. Error bars stand for 95% CI. Detailed summary statistics could be found in Supplementary Data 6.

B2) from HGI release 4 alpha, new data releases are available, offering an opportunity to evaluate the robustness of our prioritized list of risk factors across data releases. For the list of 55 significant and replicated exposures, 47 (85.45%) were also observed with the A2 and B2 GWAS from HGI release 4, while 40 (72.73%) were observed with those from HGI release 5, at the significance level of $p < 0.05$ (Supplementary Fig. 4, Supplementary Data 5 and 9). Moreover, since HGI A2 and B2 GWAS utilized population samples with unknown COVID-19 status as the control, we performed additional sensitivity analysis using HGI A1 and B1 GWAS, which compared very severe respiratory and hospitalized COVID-19 patients, respectively, with non-hospitalized COVID-19 patients as the control. Despite much smaller control-sample sizes, MR analysis with A1 and B1 still observed 36 (65.45%) of the 55 prioritized factors (Supplementary Data 5 and 10). These results suggest that our prioritized list of risk factors is robust to different data releases and control samples.

The availability of multiple phenotype definitions, including very severe respiratory COVID-19 (HGI A2), hospitalized COVID-19 (HGI B2), and any reported SARS-CoV-2 infection (HGI C2), also offers a possibility of distinguishing risk factors that affect the susceptibility to the SARS-CoV-2 infection from those that affect COVID-19 disease progression upon infection. LDSC analysis across these GWAS of different phenotype definitions and data releases revealed very high pair-wise genetic correlations, with the lowest $r_g$ of 0.85 between B2 and C2 (Supplementary Data 11). For the 55 significant and replicated exposures prioritized in our primary analysis with A2 and B2, 39 were identified ($p < 0.05$) in MR analysis with at least one C2 GWAS from releases 4 and 5, suggesting that these factors may affect both susceptibility and severity (Supplementary Fig. 4 and Supplementary Data 12 and 13). Notably, BMI-related traits and fat mass fall in this category. There are 16 significant and replicated exposures that were not identified in any MR analysis with HGI C2 GWAS, suggesting that they may mainly affect disease severity (Supplementary Data 13). For example, zinc-alpha-2-glycoprotein is associated with an increased risk of severe or hospitalized COVID-19 in our primary analysis with HGI A2 (OR: 1.37, 95% CI: 1.14–1.66), HGI B2 (OR: 1.24, 95% CI: 1.07–1.45), and NEJM (OR: 1.40, 95% CI: 1.08–1.82). But it has

no association with the risk of SARS-CoV-2 infection in analysis with HGI C2 from either release 4 (OR: 1.04, 95% CI: 0.93–1.16) or release 5 (OR: 1.00, 95% CI: 0.91–1.09). Other notable severity-related risk factors in this category include prostate-associated microseminoprotein, resistin, corneodesmosin, bradykinin, and C–C motif chemokine 23. On the other hand, there are 24 exposures that are significant ($p < 0.05$) in all three MR analyses with C2 GWAS, but not significant in our three primary MR analyses of severe COVID-19, suggesting that they may mainly affect susceptibility (Supplementary Data 13). Notable factors in this category include rheumatoid arthritis, phospholipids in small VLDL, the concentration of small VLDL particles, and matrix metalloproteinase-9. Comparisons across different COVID-19 phenotypes suggest risk factors that may mainly affect susceptibility, severity, or both. However, due to the complexity of COVID-19 disease progression and the bias towards severe cases in most COVID-19 GWAS, we would like to emphasize the preliminary nature of this analysis and caution against over-interpretation.

## Discussion

This large-scale MR study examined an extensive list of risk factors and prioritized those that potentially play causal roles in the development of severe COVID-19. It leveraged GWAS of COVID-19 of the largest sample size, and the findings were replicated with one, and for some associations, two additional COVID-19 GWAS. Using univariable MR, we first confirmed that BMI-related traits are putative causal risk factors for severe COVID-19. Our multivariable MR results further suggested that the effects of BMI-related traits might be driven by fat mass but not fat-free mass. Moreover, our findings indicate that white blood cell traits, particularly neutrophils, are inversely associated with the severe COVID-19 risk. We also highlighted scores of circulating proteins that could potentially serve as drug targets.

Our main finding that higher BMI-related traits are associated with a higher risk of severe COVID-19 is consistent with several recent MR studies[18,19,23,24]. In terms of effect size, we found that one-SD increase of BMI was causally associated with a higher risk of very severe respiratory COVID-19 (OR: 1.89, 95% CI:

1.51–2.37) and hospitalized COVID-19 (OR: 1.69, 95% CI: 1.45–1.97). In an observational study, Hamer *et al.* reported that increasing BMI had a significantly increased risk of COVID-19 hospitalization in overweight subjects (OR: 1.39, 95% CI: 1.13–1.71, 1.39), obese stage I (OR: 1.70, 95% CI: 1.34–2.16), and stage II (OR: 3.38, 95% CI: 2.60–4.40)[50]. Our MR estimates are consistent with observational estimates, while the minor differences may reflect different phenotype definitions and the impacts of confounding factors in observational associations. Furthermore, our multivariable MR analysis further showed that fat mass is a causal risk factor for severe COVID-19, while fat-free mass is not. These results suggest that the causal effect of BMI on severe COVID-19 is likely driven by fat mass. The causal effects of BMI and fat mass have plausible biological mechanisms. Fat mass has been known to have deleterious effects on lung function, inflammation, and immunity[51–53]. In adipose tissue, high production of circulating proinflammatory cytokines and adipokines may intensify virally induced inflammation and immune dysregulation, and contribute to acute respiratory distress syndrome, which is the leading cause of mortality from COVID-19[54–57]. Notably, other causal risk factors of severe COVID-19 identified in this study are also related to adiposity, including glucosamine, resistin, prostate-associated microseminoprotein, and zinc-alpha-2-glycoprotein[58–62]. These connections suggest a shared mechanism for their contribution to severe COVID-19. Therefore, further mechanistic understanding of fat mass and other related risk factors will shed light on the etiology of severe COVID-19 and provide multiple targets of intervention for prevention and treatment.

Our study indicates that white blood cell traits, especially neutrophils, are inversely associated with the risk of severe COVID-19. In addition to direct evidence from neutrophils, sum basophil–neutrophil counts and sum neutrophil-eosinophil counts are directly related to neutrophils, and concordant causal effects were obtained using multiple MR methods. We also identified myeloid white blood cell counts and granulocyte counts as being inversely associated with the risk of severe COVID-19, which is consistent with our previous MR findings[63]. In contrast to the negative associations in this MR study, previous observational studies have provided strong evidence that elevated white blood cells and neutrophils but depleted lymphocytes are common in COVID-19 patients[64–67]. This discrepancy highlights the possibility that observed associations are due to confounding and reverse causation. The causal role of neutrophils in preventing the development of severe COVID-19 has biological support. Neutrophils, the integral components of the innate immune system, are the first line of defense against invading pathogens[68]. Moreover, neutrophils participate in elaborate cell-signaling networks involving cytokines, chemokines, survival, and growth factors that cause downstream proinflammatory effects[69]. On the other hand, neutrophils are involved in the hyperinflammatory responses (e.g., overproduction of neutrophil extracellular traps and cytokine storm) in severe COVID-19 patients. This reflects the reverse causal effect of COVID-19 on neutrophils[70,71]. Overall, our present results support the potential causal effects of white blood cells, especially neutrophils, on severe COVID-19, likely through an enhanced immune response that suppresses virus infection in the early stage.

To identify potential drug targets, we found that five immune-related proteins are inversely associated with the risk of severe COVID-19. Interleukin-3-receptor subunit alpha plays important functions in hemopoietic, vascular, and immune systems[72]. Prostate-associated microseminoprotein may influence inflammation and cancer development[73]. C–C motif chemokine 23 is a chemotactic agent, which probably plays an important role in inflammation and atherosclerosis[74]. Collectin-10 can act as a cellular chemoattractant

in vitro, probably involved in the regulation of cell migration[75]. Reticulon-4 receptor influences the central nervous system and protects motoneurons against apoptosis[76]. The identification of these immune-related circulating proteins highlights the critical role of immune responses in the development of severe COVID-19.

Our study identified another six circulating proteins to be positively associated with an increased risk of severe COVID-19. Most of them are glycoproteins. The effect of zinc-alpha-2-glycoprotein might be mediated by the depletion of fatty acids from adipose tissues[77]. C1GALT1-specific chaperone 1 might abolish a glycosyltransferase function and disrupt the O-glycan Core-1 synthesis[78,79]. Corneodesmosin is a glycoprotein expressed in the epidermis and the inner root sheath of hair follicles[80]. Inter-alpha-trypsin-inhibitor heavy chain H1 is involved in cell adhesion and leukocyte migration in inflammation sites[81]. The alpha-2-macroglobulin receptor-associated protein is responsible for the role of exotoxin A in pseudomonas disease and immunity[82]. Resistin is known as a hormone that potentially links obesity to diabetes through resisting insulin action[62]. More in-depth mechanistic work is needed to better understand the physiological and biological processes through which these druggable proteins contribute to COVID-19 severity. While our study identified scores of circulating proteins, we cannot rule out the possibility that there are more COVID-19-related proteins. The number of genetic instruments is often limited for circulating proteins, precluding many of them from being analyzed. Our findings of circulating proteins not only suggest possible etiological processes but also provide potential druggable targets.

Our study has many strengths. One strength as an MR study is the ability to assess causal effects, avoiding bias from reverse causation and residual confounding. A major feature and strength of our study is an unbiased and large-scale approach to screen an extensive list of risk factors. To address the issue of multiple testing, we used FDR corrections in the discovery analysis. To ensure robustness and reduce false positives, we only reported results that were replicated in at least one replication analysis. Another strength is that we applied multivariable MR analyses to evaluate the independent causal effects of fat mass and fat-free mass. To ensure reproducibility and to encourage open science, we have released our computational pipeline. As a demonstration of the readiness of this pipeline, in the manuscript revision phage, we included additional analysis based on GWAS from HGI releases 4 and 5. We encourage readers to use our pipeline to update results for future HGI releases or for new risk factors from the IEU OpenGWAS database. Comparing our primary analysis based on HGI release 4 alpha to results from releases 4 and 5 demonstrates the high robustness of our prioritized list of risk factors. Additionally, comparing our primary analysis based on severe forms of COVID-19 to results based on any SARS-CoV-2 infection distinguishes risk factors that may mainly impact the susceptibility to infection, the severity of disease progression, or both processes.

Our study also has several weaknesses. First, our two-sample MR analysis inherits many of the limitations and complications of the COVID-19 GWAS, which meta-analyzed studies with very heterogeneous designs, ascertainment and phenotyping of cases, and sources of control samples. Although significant efforts have been spent in enhancing the robustness of genetic findings and many loci have been shown to be biologically meaningful[27], the usage of nonrepresentative convenience samples may induce collider bias that distorts phenotypic and genetic associations[28–30]. Second, although we applied multiple sensitivity analyses in our MR analysis, including the heterogeneity test, MR-Egger, and WM method, we could not fully rule out the possibility that some genetic variants might be pleiotropic. Another approach to test the assumptions of independence and no horizontal pleiotropy is to examine known associations with

potential confounders or other phenotypes using databases such as PhenoScanner[47,48]. We performed an exemplary lookup and validation for six exposures that have a small number of genetic instruments (the five significant and doubly replicated circulating proteins in Supplementary Fig. 4 and glucosamine supplementation in Supplementary Fig. 3). No associations with confounders were observed, but four exposures have instruments associated with either blood cell or BMI-related traits, suggesting potential horizontal pleiotropy (Supplementary Data 14). After excluding these genetic instruments, the effects of three exposures remain significant ($p < 0.05$, Supplementary Data 15). Therefore, this manual examination of six exposures with PhenoScanner supports the effects of five, while that of interleukin-3 receptor subunit alpha may be explained by horizontal pleiotropy. Similar in-depth analysis could be performed in follow-up studies on other exposures of interest. Third, another limitation of our study is that some GWAS of exposures and the HGI GWAS of COVID-19 have overlapped samples, especially those from the UK Biobank. This sample overlap may induce bias in MR estimates. Strong instruments for the instrument-exposure association are less susceptible to this bias and all traits included in our analysis have strong instruments ($F > 10$). To mitigate this issue, we also utilized another GWAS of COVID-19, the NEJM study, which does not have overlapping samples with exposure GWAS. Fourth, the three COVID-19 GWAS used in our primary analysis, the HGI A2, HGI B2, and NEJM study, have overlapped cases and therefore do not represent independent replications. Their inclusion in our study design may help reduce false positives in our final list of prioritized risk factors.

Fifth, the HGI A2, HGI B2, and NEJM GWAS utilized population controls without known COVID-19 status. While this practice increases statistical power with much larger control-sample sizes, it may also introduce biases in case ascertainment and thus genetic associations. It is somewhat reassuring that our sensitivity analysis with the HGI A1 and B1 GWAS, which utilized nonhospitalized COVID-19 patients as the control, also observed most of the prioritized risk factors. Sixth, while our primary analysis focuses on GWAS of severe COVID-19, the prioritized risk factors may increase the susceptibility to viral infection or exacerbate disease progression upon infection. Our attempt to distinguish susceptibility-related from severity-related risk factors by including analysis with GWAS of any COVID-19 (HGI C2) is complicated by the fact that the case definition for any COVID-19 or SARS-CoV-2 infection is biased toward severe cases. Further mechanistic research is required to decipher the biological pathways underpinning the effects of these prioritized risk factors on COVID-19. Seventh, a further weakness is that the statistical power for some exposures was limited, and some null results might be false negatives. Further positive findings may be revealed if more GWAS with larger sample sizes become available. Eighth, as an effort to reduce complications from population stratification, our study focuses on European ancestry, and thus the findings may not be generalizable to other ethnicities.

In conclusion, the present study provides evidence that the potential causal association between BMI-related traits and severe COVID-19 is driven by fat mass, but not by fat-free mass. Our findings suggest that neutrophils, granulocytes, and myeloid white blood cells are inversely associated with the severe COVID-19 risk. Our study also identifies putatively causal associations between scores of circulating proteins and severe COVID-19. These findings provide valuable insights into the etiology of severe COVID-19. These prioritized risk and protective factors could potentially serve as drug targets and guide the effective protection of high-risk populations.

## Data availability

Source data for Figs. 2 and 3 could be found in Supplementary Data 8, and that for Fig. 4 in Supplementary Data 6. All analyses were conducted using publicly available data. The exposure data (GWAS summary statistics) used in the analyses described here are freely accessible in the MR-Base platform (https://www.mrbase.org/) and the IEU OpenGWAS database (https://gwas.mrcieu.ac.uk/). Unique identifier for each dataset is available in Supplementary Data 3. We downloaded COVID-19 data (GWAS summary statistics) in the COVID-19 Host Genetics Initiative (https://www.covid19hg.org/) and the COVID GWAS results browser (https://ikmb.shinyapps.io/COVID-19_GWAS_Browser/).

## Code availability

The codes used in the Mendelian randomization analyses described here are freely accessible in TwoSampleMR R package via GitHub (https://github.com/MRCIEU/TwoSampleMR/). Full documentation for the R package is also provided (https://mrcieu.github.io/TwoSampleMR/). We implemented the MVMR analysis using the MVMR R package (https://github.com/WSpiller/MVMR/). The Mendelian randomization pipeline is available on GitHub (https://github.com/yitangsun/MR_all_COVID_19) and also deposited on Zenodo[83].

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

## Acknowledgements

We thank the investigators of the COVID-19 genome-wide association study, the COVID-19 Host Genetics Intiative, and FinnGen consortium for sharing summary-level data. We would like to express our gratitude to all other Ye lab members for stimulating discussions. K.Y. is supported by the University of Georgia Research Foundation. Funding sources had no involvement in the conception, design, analysis, or presentation of this work.

## Author contributions

Y.S. and K.Y. conceived the study. Y.S. performed data analysis and prepared visualizations. Y.S., J.Z., and K.Y. interpreted the results. Y.S. and K.Y. wrote the first draft of the paper. All authors reviewed and approved the final version.

## Competing interests

The authors declare no competing interests.

## Additional information

**Peer-review information** *Communications* Medicine thanks the anonymous reviewers for their contribution to the peer review of this work. Peer-reviewer reports are available.

