## [Peer Review File · Communications Medicine]

Reviewers' comments:

Reviewer #1 (Remarks to the Author):

Prioritizing causal risk factors for severe COVID-19: an exhaustive Mendelian randomization study

Sun et al

Communications Medicine

Comments for the authors:

This is obviously a contemporary area and one which is compelling for the application of Mendelian randomisation. However it is also one which is fraught with potential complications. For example, the very undertaking of MR work within the context of an infection and re. severity lends to the generation of biased generated as a result of the stratification of population based samples over groups on infected/non-infected - itself a feature associated with factors which may be coincident with severity factors. This and other aspects of the analysis of infections in an MR context present real methodological challenges and one would expect these to be discussed in a paper of this nature.

The work here looks to perform a form of screen-by-MR for risk factors which may be correlated with severe COVID-19. The authors state that this is an: "unbiased and exhaustive MR analysis" - which is in part true but does have some important limitations.

This is done using a two-sample framework and the top findings from this are taken to indicate that measures of fat mass and cell count look to be related (and by implication causally related to) severe disease. Indeed, the abstract of this work goes further to suggest that the study "... shows that fat mass and white blood cells underlie the etiology of severe COVID-19". Whilst some of these effect are both plausible and indeed may be true in terms of the relationship between variables of note and severe disease, the suggested and rather complete feel for the aetiological explanation of severe disease is alarming. There does not appear to be any consideration of the likely and relatively straightforward problem that when considering that risk factors for infection can be the same as those being suggested here to be important for the severity of disease. To this end, it may well be that the results here could be in some respects telling the story of the chances of infection alone or worse, may be generating biased estimates of impact on severity.

The authors do state a concentration on the assessment and attempted handling of pleiotropy. This is good to hear, though with the methods used and the genetic architecture of the traits being examined, this is arguably not the main issue in this case. The actual structure of the GWAS data sets and the nature of the estimates used in the MR are likely to be more important in this case. In brief, the problem here stems from the nature of the COVID-19 GWAS - which are brilliant, but are specific. These results can suffer from being distal examinations of infection exposure, can be subject to the problem of selection bias in the actual samples available (ie. with COVID-19 data and GWAS data) and also can suffer problems relating to the undertaking of a specific case type analysis. Importantly for this paper is the first of these. The analysis of the A2 sample and then B2 (which is closer to the NEJM analysis used) is an interesting start point here, but it is very difficult to make the distinction that this is reporting specific signals for severe disease over the experience of mild

disease or just the chances of being infected. Sources of information here potentially are the comparison of A2 and B2/NEJM in efforts to locate only those loci which are associated with severe disease. Alternatively, the incorporation of C2 HGI could be considered - though this is, of course, subject to the biased collection of cases of COVID-19 towards those symptomatic or severely ill. These analyses are all potential sub-optimal, but there are a couple of points which are important for this manuscript. First is the notion that the comparison between COVID-19 source GWAS is critical to the interpretation of the results (with ideally an assessment of severe specific genetic associations and possibly including C2 HGI). Secondly there is the likely situation that the results here are quite difficult to distinguish between the results here being specific for severe disease, any incident cases, risk of infection or a biased representation of any of these. As a consequence, it would seem prudent to be both open to these limitations in the paper and also to tone down the interpretation of the results.

As mentioned, it may be that some of the relationships are indeed true and that the estimates here reflect underlying relationships with severe disease. The reported relationships with inflammatory pathways in particular look to be of interest and may well be part of an adiposity related chronic inflammatory story which has some relationship to bad outcomes. It would seem that the trick here would be to make better use of the different GWAS performed already and to cross reference findings from MR analysis within different COVID-19 results sets in order to gain some clarity over whether the estimates here are biased by sampling frame or not. It appears that the focus of limitations on pleiotropy, overlapping samples and the ethnic origin of the samples - whilst being important - miss this central problem for COVID-19 two-sample MR completely.

Specific comments:

- Overall the paper is written well - it is clear and there are very few errors.
- Repeatedly the results sections and interpretation sections state that results “indicate” or “show that” risk factors are “driving” risk (or odds here) for severe COVID-19. These phrases are potentially too strong and really should be written as suggestions given evidence if possible.
- It would be good to comment on the value of this approach (MR of a series of risk factors) over just the systematic comparison of differing COVID-19 GWAS - i.e. looking only at “top hits”, but from GWAS of COVID-19 where the sampling frames differ (incident events/severity/etc). Is the suggestion here that the effects seen are of use or is it just the flagging of risk factors for followup that is the main purpose of the work and - for druggable targets at least - could this have been with the GWAS alone?
- The arrangement of the studies here suggests that the two HGI studies can be used as replicates. Whilst I see there to be value in the third study in this way - the comparison of the HGI studies like this is certainly not “replication” (see comments above) and the p-value threshold approach for validating initial results is not really suitable when handled in this way - rather a comparison for qualitative difference is more appropriate - but very different.
- A description of the number, type and source of the risk factors should really come before the main results to give the reader a feeling of the scale of association analyses undertaken. By saying a FDR was done, this is rather concealed.

Reviewer #2 (Remarks to the Author):

The author used Mendelian randomization (MR) analysis in the sample and found a series of factors that may be related to the pathogenicity of the coronavirus, including multiple body mass index (BMI), changes in some circulating proteins and white blood cells, but at this time There have been a series of articles that have conducted in-depth research on related mechanisms. I don't think that publishing articles at this time will provide references for others.

Reviewer #3 (Remarks to the Author):

The authors performed a thorough two-sample MR-Analysis using published GWAS and three COVID-19 studies, one as discovery, two for replication purposes. The relevance assumption of the instruments was controlled by selecting Genome-wide significant variants, other assumptions of MR were controlled by applying sensitivity measures incl. MR-Egger. They identify a role for traits including fat-mass, levels of circulating proteins, including immune-related proteins, and white blood cell traits. The study is timely, clearly written, and highly relevant, the findings help for a better understanding of the pathomechanisms of COVID-19. However, some points need to be clarified before publication

MAJOR Comments

Your study benefits from open science and public data. Please provide the R-code of your analysis as a supplement or GitHub repository to strengthen this important movement

Please work for some important examples in more detail in addition to general approaches like the MR-Egger test whether genetic variants or other associations were reported in the literature that might confound the study (independence assumption of MR) or might result in horizontal pleiotropy.

MINOR Comments

Please report on adherence to MR-recommendations, e.g. BMJ 2018; 362
doi: <https://doi.org/10.1136/bmj.k601>

You discuss that there is a certain overlap in your studies. Please estimate roughly the degree of overlap of your studies.

Please compare the MR-estimate of important top-hits with reported observational estimates.

Please add the number of identified significant MR-estimates affecting COVID-19 at the end of the flow chart in Fig.1. Stratify the identified MR-estimates in Novel, Confirming and Contradicting previously reported MR-results.

Did the authors perform a meta-analysis of the identified effect sizes on COVID-19? Please consider especially a meta-analysis of the two confirming studies HGI-B2 and NEJM for this (excluding identification bias-winners curse of the discovery study).

Thank you for providing much detail in Suppl. Tables. Please check for rows including only NAs, e.g. found in Table S3. Please comment within the suppl. table caption, why sometimes information like

category etc. is missing.

Please provide a brief explanation of the supplemental tables and columns in the suppl. table caption.

Please provide explanations of abbreviations used in the figures.

Response to Reviewer 1 Comments:

Comments for the authors:

This is obviously a contemporary area and one which is compelling for the application of Mendelian randomisation. However it is also one which is fraught with potential complications. For example, the very undertaking of MR work within the context of an infection and re. severity lends to the generation of biased generated as a result of the stratification of population based samples over groups on infected/non-infected - itself a feature associated with factors which may be coincident with severity factors. This and other aspects of the analysis of infections in an MR context present real methodological challenges and one would expect these to be discussed in a paper of this nature.

Response: We appreciate Reviewer 1 for being supportive of the overall topic and organization of the manuscript. The critical comments are mainly directed at methodological challenges and limitations. The Reviewer made an excellent point about the differences and potential complications between severity and susceptibility. In response to this, we further performed analysis using the HGI C2 study, a GWAS meta-analysis comparing SARS-CoV-2 infected individuals and population controls. It is expected that associations in the C2 study are more reflective of susceptibility to SARS-CoV-2 infection, although it is likely that the reported infection is biased towards severe cases. In addition to our original analysis based on HGI release 4-alpha, in this revision, we have performed analysis based on HGI release 4 and release 5. The results are overall consistent across different releases. We have added one Result section describing comparisons across different COVID-19 phenotypes and different data releases. Methods and Supplementary Materials have been updated accordingly.

Even though we have included extensive analysis and comparisons to show the reproducibility of our prioritized list of risk factors, we agreed that there may still be uncontrolled biases in the MR results, and possible complications in the data analysis process and in the interpretation of results. We have added many more discussions on Limitations and Weaknesses of study.

We have also revised the overall manuscript to improve clarity and incorporate suggestions. We thank the Reviewer for helping us craft a substantially improved manuscript.

The work here looks to perform a form of screen-by-MR for risk factors which may be correlated with severe COVID-19. The authors state that this is an: “unbiased and exhaustive MR analysis” - which is in part true but does have some important limitations.

Response: We have included additional analyses and more extensive discussions on limitations. We hope these alleviate your concerns.

This is done using a two-sample framework and the top findings from this are taken to indicate that measures of fat mass and cell count look to be related (and by implication causally related to) severe disease. Indeed, the abstract of this work goes further to suggest that the study "... shows that fat mass and white blood cells underlie the etiology of severe COVID-19". Whilst some of these effects are both plausible and indeed may be true in terms of the relationship between variables of note and severe disease, the suggested and rather complete feel for the aetiological explanation of severe disease is alarming. There does not appear to be any consideration of the likely and relatively straightforward problem that when considering that risk factors for infection can be the same as those being suggested here to be important for the severity of disease. To this end, it may well be that the results here could be in some respects telling the story of the chances of infection alone or worse, may be generating biased estimates of impact on severity.

Response: We thank the Reviewer for raising these constructive points. First, we agree that while MR aims to evaluate causal relationships, the results are only suggestive and require further verification. We have toned down our conclusion in the abstract and statements throughout the manuscript. We now use expressions like "may", "might", "suggest", instead of more assertive terms. Second, the Reviewer raised an excellent point about the differences between "risk factors for infection" (i.e., susceptibility) and "for the severity of disease". To directly address this concern, we perform additional MR analysis with GWAS of any SARS-CoV-2 infection (i.e., HGI C2), which is more reflective of susceptibility to SARS-CoV-2 infection. In the Results section, we highlighted some of these new results:

"For the 55 significant and replicated exposures prioritized in our primary analysis with A2 and B2, 39 were identified ($p < 0.05$) in MR analysis with at least one C2 GWAS of reported SARS-CoV-2 infection from release 4 and 5, suggesting these factors may affect both susceptibility and severity (Supplementary Fig. 4, Supplementary Tables 19-22)..... There are 16 significant and replicated exposures that were not identified in any MR analysis with HGI C2 GWAS, suggesting that they may mainly affect disease severity (Supplementary Tables 23)..... On the other hand, there are 24 exposures that are nominally significant in all three MR analyses with C2 GWAS, but not significant in our three primary MR analyses of severe COVID-19, suggesting that they may mainly affect susceptibility (Supplementary Tables 24)."

However, we recognized the preliminary nature of this analysis and possible complications. Therefore, we emphasized that:

"However, due to the complexity of COVID-19 disease progression and the bias towards severe cases in most COVID-19 GWAS, we would like to emphasize the preliminary nature of this analysis and caution against over-interpretation."

Additional discussions of limitations related to the interpretation of results are included in the Discussion section.

The authors do state a concentration on the assessment and attempted handling of pleiotropy. This is good to hear, though with the methods used and the genetic architecture of the traits being examined, this is arguably not the main issue in this case. The actual structure of the GWAS data sets and the nature of the estimates used in the MR are likely to be more important in this case. In brief, the problem here stems from the nature of the COVID-19 GWAS - which are brilliant, but are specific. These results can suffer from being distal examinations of infection exposure, can be subject to the problem of selection bias in the actual samples available (ie. with COVID-19 data and GWAS data) and also can suffer problems relating to the undertaking of a specific case type analysis. Importantly for this paper is the first of these. The analysis of the A2 sample and then B2 (which is closer to the NEJM analysis used) is an interesting start point here, but it is very difficult to make the distinction that this is reporting specific signals for severe disease over the experience of mild disease or just the chances of being infected. Sources of information here potentially are the comparison of A2 and B2/NEJM in efforts to locate only those loci which are associated with severe disease. Alternatively, the incorporation of C2 HGI could be considered - though this is, of course, subject to the biased collection of cases of COVID-19 towards those symptomatic or severely ill. These analyses are all potential sub-optimal, but there are a couple of points which are important for this manuscript. First is the notion that the comparison between COVID-19 source GWAS is critical to the interpretation of the results (with ideally an assessment of severe specific genetic associations and possibly including C2 HGI). Secondly there is the likely situation that the results here are quite difficult to distinguish between the results here being specific for severe disease, any incident cases, risk of infection or a biased representation of any of these. As a consequence, it would seem prudent to be both open to these limitations in the paper and also to tone down the interpretation of the results.

Response: *Thank you for this insightful suggestion. As suggested, “Alternatively, the incorporation of C2 HGI could be considered”, we performed additional analysis with HGI C2. Please see our relevant response above. Moreover, we agree with the Reviewer’s concerns about “selection bias in the actual samples available”, “the biased collection of cases of COVID-19 towards those symptomatic or severely ill”, and “quite difficult to distinguish between the results”. We have included relevant discussions, such as:*

“Second, another limitation is that some GWAS of exposures and the HGI GWAS of COVID-19 have overlapped samples, especially those from the UK Biobank. This sample overlap may induce bias in MR estimates. Strong instruments for the instrument-exposure association are less susceptible to this bias and all traits included in our analysis have strong instruments ($F > 10$).”

“Third, the three COVID-19 GWAS used in our primary analysis, the HGI A2, HGI B2, and NEJM study, have overlapped cases and therefore do not represent independent replications. Their inclusion in our study design helps reduce false positives in our final list of prioritized risk factors. Fourth, the HGI A2, HGI B2, and NEJM GWAS utilized population controls without known COVID-19 status. While this practice increases

statistical power with much larger control sample sizes, it may also introduce biases in case ascertainment and thus genetic associations. Fifth, while our primary analysis focuses on GWAS of severe COVID-19, the prioritized risk factors may increase the susceptibility to viral infection or exacerbate disease progression upon infection. Our attempt to distinguish susceptibility-related from severity-related risk factors by including analysis with GWAS of any COVID-19 (HGI C2) is complicated by the fact that the case definition for any COVID-19 or SARS-CoV-2 infection is biased towards severe cases. Further mechanistic research is required to decipher the biological pathways underpinning the effects of these prioritized risk factors on COVID-19.”

We have further toned down the conclusion and summarizing statements in our manuscript.

As mentioned, it may be that some of the relationships are indeed true and that the estimates here reflect underlying relationships with severe disease. The reported relationships with inflammatory pathways in particular look to be of interest and may well be part of an adiposity related chronic inflammatory story which has some relationship to bad outcomes. It would seem that the trick here would be to make better use of the different GWAS performed already and to cross reference findings from MR analysis within different COVID-19 results sets in order to gain some clarity over whether the estimates here are biased by sampling frame or not. It appears that the focus of limitations on pleiotropy, overlapping samples and the ethnic origin of the samples - whilst being important - miss this central problem for COVID-19 two-sample MR completely.

***Response:** Thank you very much for this insightful point. As suggested, we have now “cross reference findings from MR analysis within different COVID-19 results sets”, especially HGI A2, B2, C2, and NEJM. First, the three GWAS of “severe COVID-19” reveal consistent effect estimates for almost all risk factors (for example in Table 1). Second, for the 55 exposures in our significant and replicated list, 39 were also identified with MR analysis with HGI C2. The statistical significance of these exposures across multiple COVID-19 GWAS of different phenotype definition and sampling scheme lends strong support for their roles in COVID-19, although the exact mechanisms have to be examined in future studies.*

We agreed that COVID-19 two-sample MR has challenges. We have emphasized this point at the beginning of our Discussion of limitations and weaknesses:

“The central challenge for the two-sample MR is to use different phenotypes and adjustments from cohorts with extremely heterogeneous designs, sample ascertainment, and control populations.”

Specific comments:

- Overall the paper is written well - it is clear and there are very few errors.

Response: *We appreciate this very encouraging comment!*

- Repeatedly the results sections and interpretation sections state that results “indicate” or “show that” risk factors are “driving” risk (or odds here) for severe COVID-19. These phrases are potentially too strong and really should be written as suggestions given evidence if possible.

Response: *Thank you for pointing out this point. We have weakened the tone throughout this manuscript. For example,*

“Our multivariable MR analysis further suggests that the BMI-related effect might be driven by fat mass (OR: 1.63, 1.03–2.58), but not fat-free mass (OR: 1.00, 0.61–1.66).”

“The multivariable MR analysis results suggest that the causal effects of BMI-related traits on severe COVID-19 may be mainly driven by fat mass.”

- It would be good to comment on the value of this approach (MR of a series of risk factors) over just the systematic comparison of differing COVID-19 GWAS - i.e. looking only at “top hits”, but from GWAS of COVID-19 where the sampling frames differ (incident events/severity/etc). Is the suggestion here that the effects seen are of use or is it just the flagging of risk factors for followup that is the main purpose of the work and - for druggable targets at least - could this have been with the GWAS alone?

Response: *Thank you very much for raising this interesting point. GWAS only detects genetic variants associated with COVID-19. Its locations of target genes are only suggestive.*

MR analysis leverages information from GWAS but examines the effects of physical measurements, circulating biomarkers, or other diseases on COVID-19. Some of the traits, especially circulating biomarkers, are informative about the biological processes underlying COVID-19. MR analysis and GWAS are complementary approaches that each offers novel insights into the disease etiology.

- The arrangement of the studies here suggests that the two HGI studies can be used as replicates. Whilst I see there to be value in the third study in this way - the comparison of the HGI studies like this is certainly not “replication” (see comments above) and the p-value threshold approach for validating initial results is not really suitable when handled in this way - rather a comparison for qualitative difference is more appropriate - but very different.

Response: *Thank you for raising this point. We agree with the Reviewer that different HGI studies are not independent replications because they have overlapping samples. There are different possible ways of comparing results across three different analyses (A2, B2, and NEJM), including comparisons for qualitative consistency, as the review suggested. Our approach indeed has required that the effect estimates have consistent*

directions in analyses with A2, B2, and NEJM, we additionally required that the p-value is at least nominally significant ($p < 0.05$). This approach will reduce false positives in the final report of risk factors.

In the Introduction and Discussion section, we further emphasize that B2 and NEJM could not be considered as an independent replication for A2 because there are overlapping samples.

“We note that due to sample overlap and different phenotypic definitions, the HGI B2 and NEJM studies are not independent or strict replications of HGI A2. They mainly serve the purpose of reducing false positives in our prioritized list of risk factors.”

“Third, the three COVID-19 GWAS used in our primary analysis, the HGI A2, HGI B2, and NEJM study, have overlapped cases and therefore do not represent independent replications. Their inclusion in our study design helps reduce false positives in our final list of prioritized risk factors.”

- A description of the number, type and source of the risk factors should really come before the main results to give the reader a feeling of the scale of association analyses undertaken. By saying a FDR was done, this is rather concealed.

Response: Thank you for pointing out this issue. In the first paragraph of the Results section, we were trying to briefly summarize the risk factors in the study before providing further details. This whole paragraph has been updated for clarity.

“Starting with the 34,519 GWAS compiled by the IEU OpenGWAS project, we focused on the 14,385 GWAS that.....exposures with fewer instruments were excluded..... We excluded all exposures with indications of pleiotropy in their genetic instruments..... We retained 1,817 exposure GWAS for the discovery analysis with the HGI A2 study, 1,740 for the replication analysis with the HGI B2 study, and 1,915 for the replication analysis with the NEJM study (Supplementary Tables 5-10)..... The false discovery rate (FDR) approach was utilized in each MR analysis to correct for multiple testing of many exposures and to reduce false positives (Supplementary Tables 7-9).”

Response to Reviewer 2 Comments:

The author used Mendelian randomization (MR) analysis in the sample and found a series of factors that may be related to the pathogenicity of the coronavirus, including multiple body mass index (BMI), changes in some circulating proteins and white blood cells, but at this time There have been a series of articles that have conducted in-depth research on related mechanisms. I don't think that publishing articles at this time will provide references for others.

Response: *We sincerely appreciate the reviewer's candid comments, which we respectfully disagree on. At the early stage of our study, we have performed a thorough review of existing MR studies on COVID-19 and summarized the state of knowledge in Supplementary Table 1. In this revised manuscript, we explicitly added the number of novel discoveries in our study in Figure 1. Among the 55 exposures in our prioritized list, only four have been reported before. Our study is the first to include an extensive list of exposures, revealing many novel risk factors. We are also the first to perform multivariable MR analysis to examine the independent causal effects of fat mass and fat-free mass. Our study will be a milestone for any future MR study on COVID-19 and provides a list of candidate risk factors for follow-up mechanistic and interventional studies.*

Response to Reviewer 3 Comments:

The authors performed a thorough two-sample MR-Analysis using published GWAS and three COVID-19 studies, one as discovery, two for replication purposes. The relevance assumption of the instruments was controlled by selecting Genome-wide significant variants, other assumptions of MR were controlled by applying sensitivity measures incl. MR-Egger. They identify a role for traits including fat-mass, levels of circulating proteins, including immune-related proteins, and white blood cell traits. The study is timely, clearly written, and highly relevant, the findings help for a better understanding of the pathomechanisms of COVID-19. However, some points need to be clarified before publication

Response: We appreciate the reviewer's enthusiasm for the manuscript and recognition that our study is "timely, clearly written, and highly relevant, and the findings help for a better understanding of the pathomechanisms of COVID-19".

MAJOR Comments

Your study benefits from open science and public data. Please provide the R-code of your analysis as a supplement or GitHub repository to strengthen this important movement

Response: Thank you very much for this suggestion. We have provided the R code in the Code availability section.

"The Mendelian randomization pipeline is available on GitHub (https://github.com/yitangsun/MR_all_COVID_19)."

In Discussion, we also emphasized that

"To ensure reproducibility and to encourage open science, we have released our computational pipeline. As a demonstration of the readiness of this pipeline, in the manuscript revision phase, we included additional analysis based on GWAS from HGI release 4 and 5. We encourage readers to use our pipeline to update results for future HGI releases or for new risk factors from the IEU OpenGWAS database."

Please work for some important examples in more detail in addition to general approaches like the MR-Egger test whether genetic variants or other associations were reported in the literature that might confound the study (independence assumption of MR) or might result in horizontal pleiotropy.

Response: Thank you for this suggestion. We have added Supplementary Figures 1, 2 and 3 for a list of representative significant risk factors, including BMI, circulating proteins, and glucosamine supplement. These figures represent close-up examinations of outliers or other possible violation of MR assumptions.

MINOR Comments

Please report on adherence to MR-recommendations, e.g. BMJ 2018; 362 doi: <https://doi.org/10.1136/bmj.k601>;

Response: *Thank you very much for the helpful suggestion. We agreed that it would be useful to evaluate the quality of the presented evidence. We have added a STROBE (Strengthening the Reporting of Observational studies in Epidemiology)-MR checklist in the Results section.*

“This study is reported as per the guidelines for strengthening the reporting of Mendelian randomization studies (STROBE-MR, Supplementary Table 2)³¹”

31 Davey Smith, G. *et al.* STROBE-MR: Guidelines for strengthening the reporting of Mendelian randomization studies. *PeerJ Preprints* 7:e27857v1, doi:10.7287/peerj.preprints.27857v1 (2019).

You discuss that there is a certain overlap in your studies. Please estimate roughly the degree of overlap of your studies.

Response: *Thank you very much for this insightful suggestion. It is challenging to estimate sample overlaps between exposure GWAS and outcome GWAS because of the large number of exposure GWAS in our study. Instead, we have provided the F statistic for every exposure as a measure of the strength of genetic instrument, and emphasized that:*

“This sample overlap may induce bias in MR estimates. Strong instruments for the instrument-exposure association are less susceptible to this bias and all traits included in our analysis have strong instruments ($F > 10$).”

Please compare the MR-estimate of important top-hits with reported observational estimates.

Response: *Thank you very much for the helpful suggestion. We have added relevant discussions in the Discussion section, as follows:*

*“In terms of effect size, we found 1 SD increase of BMI was causally associated with a higher risk of very severe respiratory COVID-19 (OR: 1.89, 95% CI: 1.51–2.37) and hospitalized COVID-19 (OR: 1.69, 95% CI: 1.45–1.97). In an observational study, Hamer *et al.* reported increasing BMI had a significantly increased risk of COVID-19 hospitalization in overweight subjects (OR: 1.39, 95% CI: 1.13–1.71, 1.39), obese stage I (OR: 1.70, 95% CI: 1.34–2.16), and stage II (OR: 3.38, 95% CI: 2.60–4.40)³⁶. Our MR estimates are consistent with observational estimates, while the minor differences may reflect different phenotype definitions and the impacts of confounding factors in observational associations.”*

36 Hamer, M., Gale, C. R., Kivimaki, M. & Batty, G. D. Overweight, obesity, and risk of hospitalization for COVID-19: A community-based cohort study of adults in the United Kingdom. *Proc Natl Acad Sci U S A* 117, 21011-21013, doi:10.1073/pnas.2011086117 (2020).

Please add the number of identified significant MR-estimates affecting COVID-19 at the end of the flow chart in Fig.1. Stratify the identified MR-estimates in Novel, Confirming and Contradicting previously reported MR-results.

Response: *Thank you very much for the helpful suggestion. We have updated Figure 1 to reflect the reviewer's suggestion.*

Did the authors perform a meta-analysis of the identified effect sizes on COVID-19? Please consider especially a meta-analysis of the two confirming studies HGI-B2 and NEJM for this (excluding identification bias-winners curse of the discovery study).

Response: *Thank you very much for this insightful question. However, HGI-A2 does include NEJM. In addition to our original analysis based on HGI release 4-alpha, in this revision, we have performed the analysis based on HGI releases 4 and 5. The replicated studies might neutralize the winner's curse. The results are overall consistent across different releases. We have updated our Results and Supplementary Materials accordingly.*

Thank you for providing much detail in Suppl. Tables. Please check for rows including only NAs, e.g. found in Table S3. Please comment within the suppl. table caption, why sometimes information like category etc. is missing.

Response: *Thank you for catching this issue, not filtering out entries without information. We have updated Supplementary Table accordingly (Table S3 is now S4).*

Please provide a brief explanation of the supplemental tables and columns in the suppl. table caption.

Response: *Thank you for pointing out this problem. We have added a second row for a brief explanation of the supplemental tables.*

Please provide explanations of abbreviations used in the figures.

Response: *Thank you for pointing out this issue and we have updated figure legends accordingly.*

Reviewers' comments:

Reviewer #1 (Remarks to the Author):

Prioritizing causal risk factors for severe COVID-19: an exhaustive Mendelian randomization study

Sun et al

Communications Medicine

Comments for the authors:

Comments look to have been addressed reasonably well. Most of the limitations to the work have been addressed in the main text and I am relatively happy with this draft as it stands, however there are a series of specific points that still need addressing. The key thing still remains the nature of the COVID-19 GWAS and the interpretation of the betas from this and hence the interpretation of the MR.

Specific points:

- The title should really be changed to reflect the more conservative nature of the interpretation of results - perhaps "Identifying potential risk factors...."
- There remain statements which are too strong. For instance - in the abstract "Our study suggests that fat mass and white blood cells may underlie the etiology of severe COVID-19". These factors may be involved in the aetiology of disease and indeed may have evidence for that here, but they are likely to be sole factors in the presentation of disease type.
- In order to acknowledge the possible limitations to the work (against the possible gains), it would be good to note that MR is part of a possible evidence source in the introduction. Further, this section should explicitly acknowledge the particular limitations in the context of (i) an infectious disease (where it is very difficult to distinguish signals for susceptibility from risk of infection) and (ii) the case only/severity analysis problems. If this is presented early, then the reader will be able to interpret the contribution of the findings here in an appropriate context.
- The analysis is broad, but not "exhaustive"
- It is ok to state "all" GWAS in the catalogue given their variable properties will be reflected in the MR results - though there could be a comment as to how many of these are (or are not) overlapping
- I will concede this to the authors. However, please update the wording around the COVID-19 GWAS
- these are not independent results and as this should be indicated early - likely by changing the following - "The associations between genetic instruments and the risk of severe COVID-19 were evaluated based on three GWAS of COVID-19".
- On results presentation - it is currently difficult to interpret the MR effect estimates in the results section text as there is no reference to the unitary change in the exposures. For example "Specifically, suggestive associations were determined for neutrophil count (OR: 0.76, 95% CI: 0.61–0.94, p = 0.013)" - is this OR for a unit change in count? This is relevant for all exposures as I can tell

(there is mention of SD for some in the discussion).

- Comparisons across HGI releases are welcome. However this does not assess the “reproducibility” of results. Further - these analyses do not reduce false positives necessarily (many will carry the same signals and comparable biases). These are sensitivity analyses alone (given the sampling frame) and it is comparison of results across definitions and case/case-control/definition that are informative. As corrected for “replication” in version one of this paper, this should also be adjusted to match opening comments.

- As suggested in opening comments - please address the likely problems of collider bias when looking at care only analyses. This is where the comparison of results to case/control results is informative and can be used to help understand results (for example using COVID-19 GWAS results which are specific for severity rather than overlapping risk and severity in MR analyses. This is not really covered at the moment.

Reviewer #3 (Remarks to the Author):

Thank you very much for addressing most of my comments!
Here is a minor issues I would like to ask:

You addressed the previous 2nd Major point "Please work for some important examples in more detail in addition to general approaches like the MR-Egger test whether genetic variants or other associations were reported in the literature that might confound the study (independence assumption of MR) or might result in horizontal pleiotropy." by adding supply. Fig. 1 to 3.
Please add for these included three examples, whether there is any reported association in the literature of the included genetic variants that might result in violation of the independence assumption or horizontal pleiotropy. This gives an exemplary lookup in detail and exemplary validation of applied instruments.

Format of this document

Reviewers' comments are in black fonts.

Our responses are in light blue fonts and in italics.

“Excerpts from the revised manuscript are in double quotes, black fonts and italics, with new revisions highlighted in yellow background”

#####

Reviewers' comments:

Reviewer #1 (Remarks to the Author):

Prioritizing causal risk factors for severe COVID-19: an exhaustive Mendelian randomization study

Sun et al

Communications Medicine

Comments for the authors:

Comments look to have been addressed reasonably well. Most of the limitations to the work have been addressed in the main text and I am relatively happy with this draft as it stands, however there are a series of specific points that still need addressing. The key thing still remains the nature of the COVID-19 GWAS and the interpretation of the betas from this and hence the interpretation of the MR.

***Response:** We appreciate Reviewer 1 for the thorough peer-review and thoughtful comments. We are pleased that our prior revisions have addressed most of the concerns. We have taken into account the new suggestions and made further careful revisions, including additional analysis. Hopefully, the Reviewer will agree with us that the revised manuscript has addressed all concerns.*

Specific points:

- The title should really be changed to reflect the more conservative nature of the interpretation of results - perhaps “Identifying potential risk factors....”

***Response:** Following this suggestion and another one below, we have revised the title to be “Prioritizing potential causal risk factors for severe COVID-19: an extensive Mendelian randomization study”. We added “potential” and changed “exhaustive” to “extensive” in order to weaken the statement. We used “extensive” to emphasize that our study examined an extensive list of exposures, including many existing GWAS in the IEU OpenGWAS project.*

- There remain statements which are too strong. For instance - in the abstract “Our study

suggests that fat mass and white blood cells may underlie the etiology of severe COVID-19". These factors may be involved in the aetiology of disease and indeed may have evidence for that here, but they are likely to be sole factors in the presentation of disease type.

Response: We have revised throughout the manuscript to weaken statements. For instance:

"Our study suggests that fat mass and white blood cells might be involved in the development of severe COVID-19".

"Our univariable MR analyses revealed evidence of potential causal effects for some circulating proteins."

"Our main finding that higher BMI-related traits are associated with a higher risk of severe COVID-19 is consistent with several recent MR studies"

We also added possible limitations of MR early in Introduction and additional caveats in the Discussion. We hope the Reviewer agrees with us that these revisions address the concerns of overstatement and overinterpretation of our results.

- In order to acknowledge the possible limitations to the work (against the possible gains), it would be good to note that MR is part of a possible evidence source in the introduction. Further, this section should explicitly acknowledge the particular limitations in the context of (i) an infectious disease (where it is very difficult to distinguish signals for susceptibility from risk of infection) and (ii) the case only/severity analysis problems. If this is presented early, then the reader will be able to interpret the contribution of the findings here in an appropriate context.

Response: Thank you for this suggestion. We agree that it is a good idea to explicitly acknowledge these particular limitations early in the manuscript. We have revised accordingly in the Introduction section:

"The recent release of large GWAS meta-analysis for various COVID-19 phenotypes offers a great opportunity for MR studies²⁷. However, special care and caution are also needed when interpreting results. Sampling from COVID-19 patients, individuals tested for infection, voluntary participants, or existing cohorts may result in non-representative samples and include collider bias that distorts phenotypic and genetic associations²⁸⁻³⁰. The inherent complexity of COVID-19 as an infectious disease and the potential complications in ascertaining cases and controls make it challenging to disentangle risk factors for increased chance of infection, susceptibility to infection, and disease severity^{27,28}."

- The analysis is broad, but not "exhaustive"

Response: Although our study strived to include almost all existing GWAS in the IEU OpenGWAS project, we agreed with the reviewer that it still may not be "exhaustive". There are still exposures that were not included in the database or were excluded in our analysis. We have revised our title to be "an extensive Mendelian randomization study". Moreover, we have revised the main text to remove any claim of "exhaustive".

- It is ok to state “all” GWAS in the catalogue given their variable properties will be reflected in the MR results - though there could be a comment as to how many of these are (or are not) overlapping - I will concede this to the authors. However, please update the wording around the COVID-19 GWAS - these are not independent results and as this should be indicated early - likely by changing the following - “The associations between genetic instruments and the risk of severe COVID-19 were evaluated based on three GWAS of COVID-19”.

Response: Thank you for these suggestions and for recognizing the fact that different GWAS for the same trait will be eventually reflected in the MR results. It is challenging to accurately quantify the “duplication rate” of GWAS for the same trait, since the names for the same trait could vary in different GWAS. To make it clear that our analysis included all GWAS and was performed at the level of each GWAS, we revised the Introduction as follows:

“All existing GWAS, as compiled by the Integrative Epidemiology Unit (IEU) OpenGWAS project, were included. We note that some GWAS were on the same traits. In each of these GWAS, independent genetic variants at the genome-wide significance were selected as instrumental variables for the studied trait.”

Also, to emphasize early that the three GWAS of COVID-19 are not independent, we revised the sentence as follows:

“The associations between genetic instruments and the risk of severe COVID-19 were evaluated based primarily on three non-independent GWAS of COVID-19..... We note that due to sample overlap and different phenotypic definitions, the HGI B2 and NEJM studies are not independent or strict replications of HGI A2.”

- On results presentation - it is currently difficult to interpret the MR effect estimates in the results section text as there is no reference to the unitary change in the exposures. For example “Specifically, suggestive associations were determined for neutrophil count (OR: 0.76, 95% CI: 0.61–0.94, $p = 0.013$)” - is this OR for a unit change in count? This is relevant for all exposures as I can tell (there is mention of SD for some in the discussion).

Response: Thank you for making this great point. The units of exposures were directly from the previous GWAS of the exposures. In Supplementary Table 4, we have a column (i.e., “unit”) reporting the unit for each exposure. We retrieved this information from the IEU OpenGWAS database. Note that the unit information is missing for some exposures in the database. For all exposures reported in Table 1 and the main text, we went back to the original GWAS and confirmed the reported units.

In the method section, we added clarifications that:

“...details for each exposure and its GWAS, including units of the exposure measurements, are available in Supplementary Table 4. The units of the exposures follow the definitions in the prior GWAS since we directly used the summary statistics. We note that all exposures reported in the main text and Table 1 have the unit of standard deviation.”

In the footnote of Table 1, we added:

“Odds ratios are measured per SD increment in the exposures.”

In the Results section, we explicitly wrote “OR per SD increment” when we first described a category of exposures. For instance:

“Specifically, suggestive associations were determined for neutrophil count (OR per SD increment: 0.76, 95% CI: 0.61–0.94, $p = 0.013$), sum basophil neutrophil counts (OR: 0.71, 95% CI: 0.57–0.87, $p = 0.001$)...”

- Comparisons across HGI releases are welcome. However this does not assess the “reproducibility” of results. Further - these analyses do not reduce false positives necessarily (many will carry the same signals and comparable biases). These are sensitivity analyses alone (given the sampling frame) and it is comparison of results across definitions and case/control/definition that are informative. As corrected for “replication” in version one of this paper, this should also be adjusted to match opening comments.

Response: *Thank you for this suggestion. We have revised the corresponding paragraph to remove descriptions of “reproducibility” and instead to emphasize the robustness of our results to different data releases and control samples in COVID-19 GWAS.*

“While our primary analysis utilized COVID-19 GWAS (A2 and B2) from HGI release 4 alpha, new data releases are available, offering an opportunity to evaluate the robustness of our prioritized list of risk factors across data releases. For the list of 55 significant and replicated exposures, 47 (85.45%) were also observed with the A2 and B2 GWAS from HGI release 4, while 40 (72.73%) were observed with those from HGI release 5, at the nominal significance level ($p < 0.05$, Supplementary Fig. 4, Supplementary Tables 11 and 15-18)”

- As suggested in opening comments - please address the likely problems of collider bias when looking at case only analyses. This is where the comparison of results to case/control results is informative and can be used to help understand results (for example using COVID-19 GWAS results which are specific for severity rather than overlapping risk and severity in MR analyses. This is not really covered at the moment.

Response: *Thank you for emphasizing the limitations of case only analysis (i.e., COVID-19 GWAS using population controls) and the suggestion of using additional COVID-19 GWAS using controls with better-defined COVID-19 status. In this revision, we have performed additional analysis using the HGI A1 and B1 GWAS, which respectively compared very severe respiratory and hospitalized COVID-19 patients to non-hospitalized COVID-19 patients at the control. Although with much smaller control sample sizes, MR analysis with A1 and B1 still observed 65% of our prioritized 55 risk factors, suggesting our results are relatively robust to the usage of different control samples.*

“...Moreover, since HGI A2 and B2 GWAS utilized population samples with unknown COVID-19 status as the control, we performed additional sensitivity analysis using HGI A1 and B1 GWAS, which compared very severe respiratory and hospitalized COVID-19 patients, respectively, to non-hospitalized COVID-19 patients as the control. Despite much smaller control sample sizes, MR analysis with A1 and B1 still observed 36 (65.45%) of the 55 prioritized factors (Supplementary Tables 11 and 19-21). These results suggest that our prioritized list of risk factors is robust to different data releases and control samples.”

However, even with this additional analysis, we recognized that there are still potential issues of collider bias from non-representative samples. We added additional discussions of limitations.

“Our study also has several weaknesses. First, our two-sample MR analysis inherits many of the limitations and complications of the COVID-19 GWAS, which meta-analyzed studies with very heterogeneous designs, ascertainment and phenotyping of cases, and sources of control samples. Although significant efforts have been spent in enhancing the robustness of genetic findings and many loci have been shown to be biologically meaningful²⁷, the usage of non-representative convenience samples may induce collider bias that distorts phenotypic and genetic associations²⁸⁻³⁰.”

References:

27 Initiative, C.-H. G. Mapping the human genetic architecture of COVID-19. *Nature*, doi:10.1038/s41586-021-03767-x (2021).

28 Griffith, G. J. et al. Collider bias undermines our understanding of COVID-19 disease risk and severity. *Nat Commun* **11**, 5749, doi:10.1038/s41467-020-19478-2 (2020).

29 Gkatzionis, A. & Burgess, S. Contextualizing selection bias in Mendelian randomization: how bad is it likely to be? *Int J Epidemiol* **48**, 691-701, doi:10.1093/ije/dyy202 (2019).

30 Munafò, M. R., Tilling, K., Taylor, A. E., Evans, D. M. & Davey Smith, G. Collider scope: when selection bias can substantially influence observed associations. *Int J Epidemiol* **47**, 226-235, doi:10.1093/ije/dyx206 (2018).”

Reviewer #3 (Remarks to the Author):

Thank you very much for addressing most of my comments!

Here is a minor issues I would like to ask:

You addressed the previous 2nd Major point "Please work for some important examples in more detail in addition to general approaches like the MR-Egger test whether genetic variants or other associations were reported in the literature that might confound the study (independence assumption of MR) or might result in horizontal pleiotropy." by adding supply. Fig. 1 to 3. Please add for these included three examples, whether there is any reported association in the literature of the included genetic variants that might result in violation of the independence assumption or horizontal pleiotropy. This gives an exemplary lookup in detail and exemplary validation of applied instruments.

Response: *Thank you for recognizing that our previous revisions have addressed most of your comments. We followed your suggestion here and utilized PhenoScanner to identify known associations for genetic instruments. The analysis with PhenoScanner has to be done in an*

exposure-by-exposure manual manner in order to separate horizontal from vertical pleiotropy. We performed an exemplary lookup and validation for six exposures presented in Supplementary Fig. 2 and 3. We did not perform similar analysis with BMI (Supplementary Fig 1) due to the huge number of genetic instruments. For the six exposures, we performed sensitivity analysis by excluding genetic instruments associated with either blood cell or BMI-related traits. Only one exposure was no longer significant. We have added these results and discussions in the revised manuscript:

“...Another approach to test the assumptions of independence and no horizontal pleiotropy is to examine known associations with potential confounders or other phenotypes using databases such as PhenoScanner^{72,73}. We performed an exemplary lookup and validation for six exposures that have a small number of genetic instruments (the five significant and doubly replicated circulating proteins in Supplementary Fig. S2 and glucosamine supplementation in Supplementary Fig. S3). No associations with confounders were observed, but four exposures have instruments associated with either blood cell or BMI-related traits, suggesting potential horizontal pleiotropy (Supplementary Tables 28 and 29). After excluding these genetic instruments, the effects of three exposures remain nominally significant (Supplementary Table 30). Therefore, this manual examination of six exposures with PhenoScanner supports the effects of five, while that of interleukin-3 receptor subunit alpha may be explained by horizontal pleiotropy. Similar in-depth analysis could be performed in follow-up studies on other exposures of interest...”

References:

72 Staley, J. R. et al. PhenoScanner: a database of human genotype-phenotype associations. *Bioinformatics* **32**, 3207-3209, doi:10.1093/bioinformatics/btw373 (2016).

73 Kamat, M. A. et al. PhenoScanner V2: an expanded tool for searching human genotype-phenotype associations. *Bioinformatics* **35**, 4851-4853, doi:10.1093/bioinformatics/btz469 (2019).”

REVIEWERS' COMMENTS:

Reviewer #1 (Remarks to the Author):

Many thanks for addressing points raised in my recent re-review. Most of these are now ok and bar minor comments (which are not really going to change the paper much), I am happy with the revision.

Two things that could be altered would be an avoidance of subjective terms like "nominal" to describe p value thresholds - just treat them as they are. Secondly, I would encourage the suggestion of LDSR in order to look at overlap and between studies and then also rg estimates to deconfound the interpretation of different analyses. Biases in case only analysis will remain problematic.

Thanks for being attentive.

Format of this document

Reviewers' comments are in black fonts.

Our responses are in light blue fonts and in italics.

“Excerpts from the revised manuscript are in double quotes, black fonts and italics, with new revisions highlighted in yellow background”

#####

Reviewers' comments:

Reviewer #1 (Remarks to the Author):

Many thanks for addressing points raised in my recent re-review. Most of these are now ok and bar minor comments (which are not really going to change the paper much), I am happy with the revision.

Response: We sincerely appreciate your kind comments. We are pleased that our prior revisions have addressed most of the concerns.

Two things that could be altered would be an avoidance of subjective terms like "nominal" to describe p value thresholds - just treat them as they are.

Response: Thank you very much for pointing it out. We deleted the “nominal” in our revised manuscript.

Secondly, I would encourage the suggestion of LDSR in order to look at overlap and between studies and then also rg estimates to deconfound the interpretation of different analyses. Biases in case only analysis will remain problematic.

Response: We appreciate your insightful suggestion. In this revision, we have performed genetic correlation analysis.

In the method section, we added:

“In addition, genetic correlations between various COVID-19 phenotypes from HGI releases 4 and 5 were estimated using linkage disequilibrium score (LDSC) regression^{36,37}.”

References:

36 Bulik-Sullivan, B. K. et al. LD Score regression distinguishes confounding from polygenicity in genome-wide association studies. Nat Genet 47, 291-295, doi:10.1038/ng.3211 (2015).

37 Bulik-Sullivan, B. et al. An atlas of genetic correlations across human diseases and traits. Nat Genet 47, 1236-1241, doi:10.1038/ng.3406 (2015).”

In the Results section, we described the genetic correlations among various COVID-19 phenotypes.

“LDSC analysis across these GWAS of different phenotype definitions and data releases revealed very high pair-wise genetic correlations, with the lowest r_g of 0.85 between B2 and C2 (Supplementary Data 11).”

Thanks for being attentive.

***Response:** We would like to sincerely thank Reviewer 1 for providing insightful feedback to strengthen our manuscript. We sincerely hope that our responses have addressed all your concerns.*